# Biomechanical Evaluation of Plantar Pressure Distribution towards a Customized 3D Orthotic Device: A Methodological Case Study through a Finite Element Analysis Approach

Jesus Alejandro Serrato-Pedrosa *, Guillermo Urriolagoitia-Sosa *, Beatriz Romero-Ángeles *, Guillermo Manuel Urriolagoitia-Calderón, Salvador Cruz-López, Alejandro Urriolagoitia-Luna, David Esaú Carbajal-López, Jonathan Rodolfo Guereca-Ibarra and Guadalupe Murillo-Aleman

Instituto Politécnico Nacional, Escuela Superior de Ingeniería Mecánica y Eléctrica, Sección de Estudios de Posgrado e Investigación, Unidad Profesional Adolfo López Mateos, Edificio 5, 2do, Piso, Biomechanics Group, Col. Lindavista, Del. Gustavo A. Madero, Ciudad de México 07320, Mexico; urrio332@hotmail.com (G.M.U.-C.); salvadorcruzlopezim@gmail.com (S.C.-L.); alex_ul56@hotmail.com (A.U.-L.); esaucarba99@gmail.com (D.E.C.-L.); guerecatic@gmail.com (J.R.G.-I.); guadalupe.murillo.a95@gmail.com (G.M.-A.)

* Correspondence: alejandroserrato@live.com.mx (J.A.S.-P.); guiurri@hotmail.com (G.U.-S.); romerobeatriz97@hotmail.com (B.R.-Á.)

**Abstract:** Plantar pressure distribution is a thoroughly recognized parameter for evaluating foot structure and biomechanical behavior, as it is utilized to determine musculoskeletal conditions and diagnose foot abnormalities. Experimental testing is currently being utilized to investigate static foot conditions using invasive and noninvasive techniques. These methods are usually expensive and laborious, and they lack valuable data since they only evaluate compressive forces, missing the complex stress combinations the foot undergoes while standing. The present investigation applied medical and engineering methods to predict pressure points in a healthy foot soft tissue during normal standing conditions. Thus, a well-defined three-dimensional foot biomodel was constructed to be numerically analyzed through medical imaging. Two study cases were developed through a structural finite element analysis. The first study was developed to evaluate barefoot behavior deformation and stresses occurring in the plantar region. The results from this analysis were validated through baropodometric testing. Subsequently, a customized 3D model total-contact foot orthosis was designed to redistribute peak pressures appropriately, relieving the plantar region from excessive stress. The results in the first study case successfully demonstrated the prediction of the foot sole regions more prone to suffer a pressure concentration since the values are in good agreement with experimental testing. Employing a customized insole proved to be highly advantageous in fulfilling its primary function, reducing peak pressure points substantially. The main aim of this paper was to provide more precise insights into the biomechanical behavior of foot pressure points through engineering methods oriented towards innovative assessment for absolute customization for orthotic devices.

**Keywords:** plantar pressure; foot soft tissue; finite element analysis; medical imaging; 3D foot orthosis; orthotic devices; baropodometric testing

## 1. Introduction

Over recent years, there has been an increasing trend in the medical scientific community of studying and analyzing the print of plantar pressure distribution for an optimal understanding of the biomechanics of the foot relying on its load distribution. It is a reliable parameter for analyzing foot functions and provides further insights into the studies of the etiology of several lower limb musculoskeletal problems. Within the medical field, the measurement of these loads through footprints has been used from the oldest and most traditional methods, up until the development of computerized equipment specialized

in this task, for accurately acquiring the pressure points on the foot externally [1–4]. The oldest tracing and sketching of footprint techniques, whether using any medical device or only paper and ink, has been established as a standard for the structural evaluation of the foot and body balance [5]. Innovative force platforms and pressure-sensing insoles are cutting-edge technologies for standing and dynamic pressure calculations [6–8].

Highly invasive and painful procedures are required to comprehend how these forces affect internal foot tissues. Thus, experimental tests (similar to compression tests) are usually performed on cadaveric feet, simulating their behavior under various amounts of load [9,10]. Obtaining the pressure points in the foot is considered one of the guidelines for understanding its normal and pathological function and determining stress behaviors, total displacements, total strains, and contact areas. These tests are remarkable approaches to understanding the complex foot mechanism of distributing loads within its unique capability of adapting to different ground geometries. Furthermore, pressure distribution varies from subject to subject since it is influenced by particular factors such as gender, age, race, and weight, to mention a few [11,12].

As it can be inferred from the above, a common problem for the multidisciplinary science of biomechanics is the professional equipment needed to perform studies and the fact that typically, to obtain better results, it is required to perform in vivo tests on the subject under investigation [13,14]. Employing a highly detailed segmented biomodel through medical imaging can provide an essential tool to assist the healthcare sector and biomechanics professionals and can partially replace and complement experimental testing.

Due to the increasing computational development, it has been possible to perform numerical analysis and obtain accurate and reliable estimates for various parts of the body, specifically through the finite element method (FEM) [15–17]. A 3D biological model is constructed by implementing the medical branch of imaging, which is considered a standard for obtaining complex geometries of human biological systems [18,19]. Numerical analyses are close estimations that solve complex problems utilizing partial differential equations. Such methods are forms of numerical–computational analysis, where these mathematical models are represented by a discretization of connected nodes, where a mesh-like layer covers the geometry analyzed, and the nodes are points joining the mesh. The discretization's complexity and finesse help obtain more accurate approximations of the problem's solution. Nonetheless, this requires high computational resources [20,21].

This research aims to deepen the knowledge and current perceptions of this lower extremity's biomechanical behavior to enhance the design of personalized plantar orthoses. Thus, the following research focuses on analyzing foot soft tissue behavior during normal standing conditions, obtaining the pressure points on the plantar surface, and designing a customized 3D model foot orthosis to re-evaluate pressure distributions. Likewise, this research also aims to provide relevant data on the intrinsic muscles of the foot and skin behavior under pressure and the direct effect of wearing a personalized 3D model total-contact foot orthosis.

## 2. Materials and Methods

### 2.1. Footprint Sketching

Before numerically analyzing foot soft tissue behavior to develop a 3D model insole, it is relevant to rely on sketching techniques to determine whether the foot of the participant can be considered a foot in normal conditions. Thus, it is possible to evaluate the state of the foot. Various static methods for obtaining such a footprint and analyzing foot structure exist. Indeed, these methods are advantageous due to their low cost, lack of specialized or sophisticated equipment, and ease of application. The method selected for its popularity and high reliability in foot classification criteria among biomechanical and medical researchers was the Hernández Corvo method [22]. Usually, this methodology uses sketching techniques using the photopodogram or the pedigraph. The photopodogram technique uses ink or paint to obtain the footprint when the subject steps on thermographic paper over a flat surface. Whereas the pedigraph method is very similar to the previous

one, it differs in the subject stepping on a soft, foamy rubber surface filled with ink with a sheet placed underneath it. Subsequently, to analyze the results, perpendicular lines are drawn in different coincident sections along the length and width of the rearfoot and forefoot. The distance from the metatarsal area is X, and the site from the outer arch to the midfoot bearing surface is Y. When the measured lengths are obtained, an equation is applied that yields a result in the form of a percentage, which is further weighted in a broad and complete classification of foot types (Table 1) [23].

$$\text{H.C. (\%)} = \frac{(X - Y)}{X} * 100 \tag{1}$$

**Table 1.** Foot classification according to the Hernández Corvo method [22,23].

| H. C. (%) | Foot Type |
|---|---|
| 0–34 | Flatfoot |
| 35–39 | Flatfoot–Normal |
| 40–54 | Normal |
| 55–59 | Normal–Cavus |
| 60–74 | Cavus Foot |
| 75–84 | Severe Cavus Foot |
| 85–100 | Extreme Cavus Foot |

*2.2. Biomodeling Methodology*

The process by which 2D images are processed and converted to 3D matrices that generate models is known as segmentation, transforming the pixels of 2D visualizations into volumetric pixels, isovoxels, or simply voxels, in 3D [24].

Among the various programs, Simpleware ScanIP® 3.2 Build 1 and Materialise MIMICS® Research 21.0 stand out from the rest because of their advanced tools for generating 3D models. Specifically, to obtain the model of the foot, a computed tomography (CT) scan was used on a 30-year-old Mexican young adult in apparently healthy condition with a height of 1.80 m and a weight of 80 kg who usually exercises, having a regular complexion with a 24.7 kg/m$^2$ normal-range body mass index (BMI) and a foot in normal conditions (foot length of 256 mm and forefoot width of 137 mm). The described medical imaging study was conducted utilizing a high-resolution SIEMENS SOMATOM Emotion 16-slice configuration CT scan, which provided 16 images per second with a 0.6 mm distance between slices. Once the imaging study was performed, the visualization of the DICOM images and segmentation for constructing the 3D model were performed in Simpleware ScanIP® 3.2 Build 1. The reading of the tomographic study images in the program yielded a total number of slices in the transverse (axial) plane of 357, 260 in the sagittal plane, and 454 in the coronal (frontal) plane. A total of 1071 slices in all anatomical planes were obtained. Thus, it was possible to construct a well-defined model of the foot of the participant.

The methodology employed to reconstruct the foot biomodel has been recognized as setting the guidelines in 3D biological tissue reconstruction [17,25]. The methods mentioned can be briefly described in the following points:

- Development of the medical imaging study (foot and ankle).
- Acquisition of images in DICOM format.
- Image importation into the Simpleware ScanIP® 3.2 Build 1 software.
- Determination of regions and tissues of interest (foot muscle and skin).
- Segmentation of soft tissue areas of interest through different masks.
- Implementation of smoothing tools to refine the 3D biomodel.
- Exportation of the biomodel to Materialise 3-Matic® Research 13.0 to fix any segmentation process error.
- In Materialise 3-Matic® Research 13.0, solidification of the model and application of a re-mesh to acquire uniform-size elements.

- Biological model exportation to a finite element method software to implement a numerical analysis.

With the particular research purpose of analyzing the pressure points on the sole, two soft tissue structures in the foot, skin, and muscle were modeled. There are 22 intrinsic muscles distributed in 4 different layers or volumes in a concentration of various tissues, mainly fatty tissue. Intrinsic muscles provide support and stability in the foot, in contrast to the extrinsic muscles responsible for movement and forces in the foot [26]. Therefore, it was decided to represent them as a solid encapsulated body of the total muscles. On the other hand, skin segmentation was defined as shown in the images of the imaging study.

As shown in Figure 1, the model is wholly segmented, avoiding empty pixels that could cause a subsequent failure due to a missing element. Likewise, the model is smoothed or rounded along its contour to prevent any peak or excess pixel from causing problems.

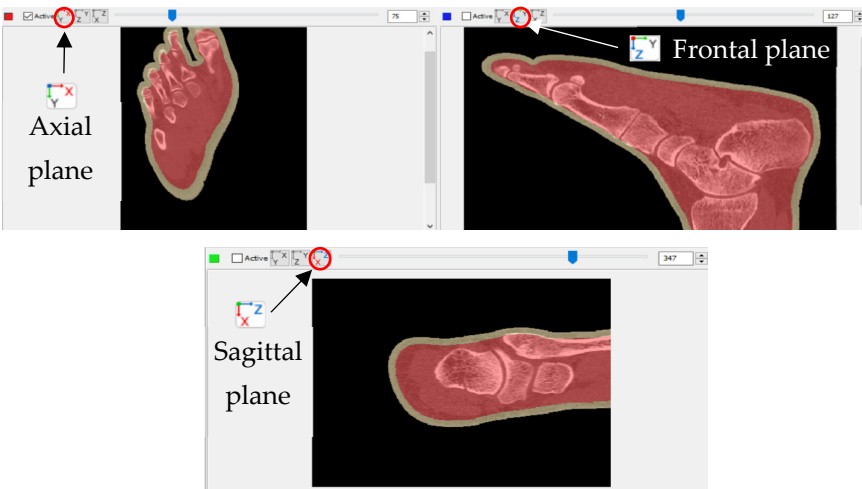

**Figure 1.** Plane views during the segmentation process. Axial plane section location corresponds to 21% of total slices. Frontal plane section location corresponds to 48.8% of total slices. Sagittal plane section location corresponds to 76.43% of total slices.

Once the model was wholly segmented and well defined, a rendering of the model was generated to smooth it and obtain more refined elements (Figure 2a). The product developed in the Simpleware ScanIP® 3.2 Build 1 software is considered a point cloud since it is hollow as a structure and not solid. Solidifying the obtained structure is mandatory to conduct a numerical analysis of the developed model. Therefore, the model generated from Simpleware ScanIP® 3.2 Build 1 was exported in an STL file extension to a computer-aided design (CAD) software capable of working with surfaces. A solidified realistic model can be achieved by refining the segmented model and closing gaps from the previous process. Notably, the design optimization software Materialise 3-Matic® Research 13.0 was used for this study to refine the surfaces to complete the model. Edges in the model were refined by using a smoothing tool. The assembly of the two solid elements (skin and muscles) and a re-meshing were added to the model to optimize its handling when numerically analyzing it (Figure 2b). Re-meshing provided uniform elements all over the complex foot geometry, allowing nodes to create a better connection among elements. This process increased the total number of nodes from 58,605 in elements with very different sizes to homogeneous elements with a total of 196,576 nodes.

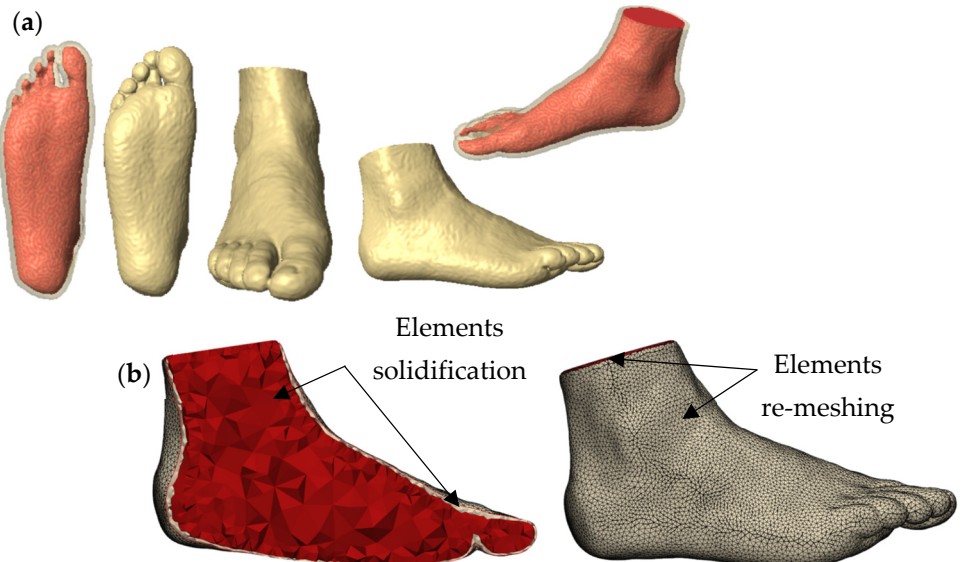

**Figure 2.** Generated biological models. (**a**) Implemented model from Simpleware ScanIP® 3.2 Build 1. (**b**) Solidification and re-meshing of the model from Materialise 3-Matic® Research 13.0.

### *2.3. Numerical Analysis of the Foot Biomodel*

### 2.3.1. First Case Study

This first numerical analysis focused on studying the pressure points in the foot sole during standing, where the foot is considered in a neutral or medium support position since it is the most fundamental anatomical position of the foot to evaluate. Thus, in this position, the foot is structurally analyzed with an external agent in compression towards the plantar surface. The upper regions of the soft tissues were represented as fixed in all degrees of freedom, embedded, to simulate the effects of the supinator tissue constraints of the ankle. Likewise, all degrees of freedom were constrained at the top of the forefoot, instep, toes, and around the foot due to the softness of the tissues. In addition, a concrete plate with a vertical displacement was used as an external agent to simulate the impact produced on the sole by ground reaction forces. A 0.6 coefficient of friction between the foot and the ground was also set [27]. Figure 3 shows the representation of a free-body diagram for the loading and boundary conditions.

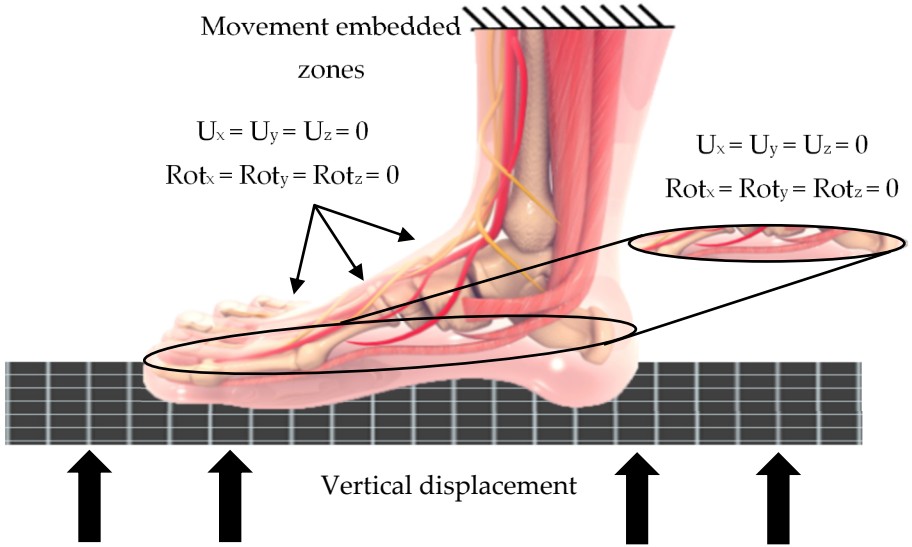

**Figure 3.** General biomodel free-body diagram. U refers to the displacement in the specified axis; Rot refers to rotation in the mentioned plane.

Soft tissues have extremely complex characteristics, being multilayer structures reinforced with collagen and with a nonlinear and anisotropic behavior, in addition to being considered hyperelastic and viscoelastic materials. For this study, the characteristics of the skin and muscle were simplified and considered with linear-elastic, continuous, homogeneous, and isotropic behavior, taking the values provided in the literature on foot biomechanical models by Luboz and Wu [28,29]. The assignment of two different mechanical properties for the muscle relied on developing an analysis with a partially conservative approach. In addition, the mechanical properties of the plate representing the ground were selected from the literature [30]. The mechanical property values can be seen in Table 2. Once the mechanical properties were assigned, the discretization process was developed using high-order 3D solid elements and generating 20 nodes per element. The analysis had three parts: skin, muscle encapsulation, and plate. A total of 371,120 elements and 196 576 nodes were obtained by fine and semi-controlled discretization (Figure 4). The discretization of the ground support was much less refined than that of the biomodel to save computational resources.

**Table 2.** Mechanical properties of the elements [28–30].

| Material | Young's Modulus (MPa) | Poisson's Ratio |
| --- | --- | --- |
| Foot skin | 0.2 | 0.485 |
| Foot muscles (Luboz) | 0.06 | 0.495 |
| Foot muscles (Wu) | 1.08 | 0.49 |
| Ground support | 210,000 | 0.3 |

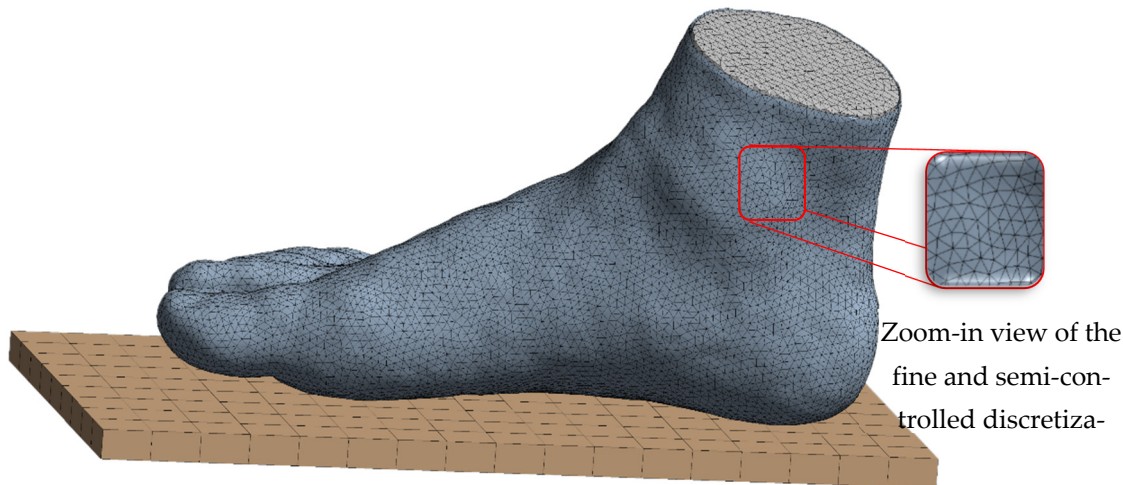

Zoom-in view of the fine and semi-controlled discretiza-

**Figure 4.** ANSYS Workbench® 2021 R2 model discretization.

Based on the established loading and boundary conditions, the upper area of the model, the forefoot, and the medial and lateral zones of the foot are also embedded. The constraint regions around the foot consist of a tape with a width of 2 mm relative to the dimensions of the modeled foot, avoiding an unreal lateral displacement when the load is applied. Likewise, the constraint in the instep and toe area has the same intention of controlling excessive vertical displacement. The external agent is assumed to be the plate, performing a vertical indentation to produce a displacement of 5 mm towards the plantar surface of the model to generate vertical loads. Since the weight of the person´s foot analyzed is 80 kg, an exerting force of 400 N is produced in each foot (Figure 5). According to experimental bases, there is a strong relationship when a force of 400 N produces a displacement magnitude of approximately 5 mm. Similarly, the foot has a constant displacement of between 4.8 and 5.6 mm while maintaining the anatomical position of balanced standing. In addition, evidence considers the application of an external agent within a displacement acting as a

pressure rather than a load since it generates estimations closer to the natural behavior of the biomechanical characteristics of the plantar surface [29–33].

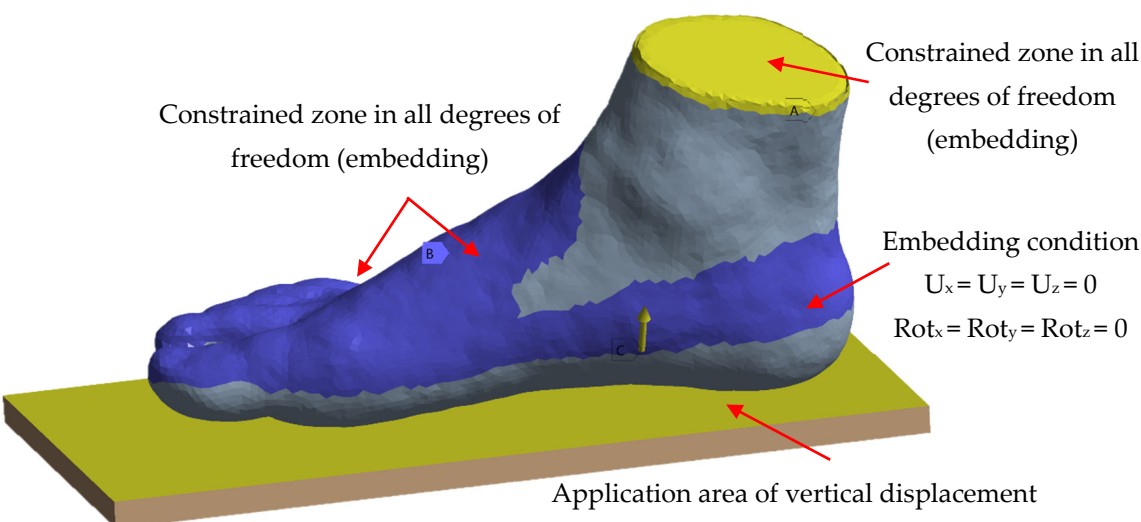

**Figure 5.** ANSYS Workbench® 2021 R2 loading and boundary condition assignment. U refers to the displacement in the specified axis; Rot refers to rotation in the mentioned plane.

2.3.2. Second Case Study

The second case study aimed to numerically evaluate the foot sole within the same anatomical position and mechanical principles, normal standing conditions. Nonetheless, a 3D personalized full-length total contact thermoplastic polyurethane (TPU) insole model was implemented between the foot sole contact points with the ground support to reduce pressure peaks. When an orthotic device is used, the insole cushioning effects absorb most ground reaction forces, and its performance is visualized in the biomechanical behavior results of the plantar region.

The employment of TPU as the insole material was due to several recent studies demonstrating its highly impressive characteristics; it has the qualities for use as an additively manufacturable material capable of being physically manufactured via fused filament fabrication (FFF), not requiring a high cost to produce and being suitable for 3D printing. Moreover, it has ideal mechanical properties for stress redistribution, compression strength support, and pain relief; in addition, it is biocompatible and sustainably advantageous for 3D printing manufacturing [34–41].

Many methodologies were reviewed to design an optimal biomodel closest to the specific right foot morphology of the participant. The refined 3D biological model developed for the numerical analysis was employed to obtain a positive foot impression from a box made in SpaceClaim® 2021 R2 CAD software, simulating a cast physically taken from an orthopedist (Figure 6a) [42,43]. The foot silhouette was sketched from the impression taken to generate the insole contour (Figure 6b). The 3D biomodel was placed right above the insole, using Boolean operations and working with surfaces to create a customized insole based on parametric designs set by specialized insole design software [44–47]. Working with surface modeling for the insole design allowed certain regions to be smoothed and the orthosis to be adjusted to foot morphology (Figure 6c). Factors such as total length, heel and toe thickness, width, and draft angle were considered. The insole has a 3 mm thickness, which is not a crucial factor for this case study evaluating the force distributions during regular standing since loads below 800 N are unimportant for 3D-printed devices made from TPU [48].

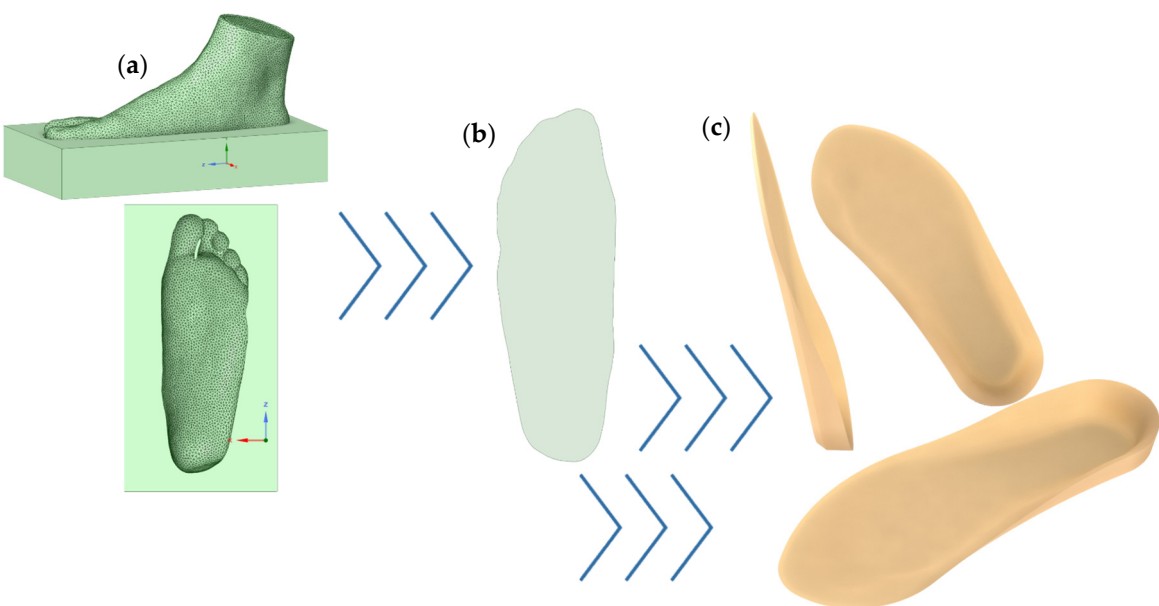

**Figure 6.** Process of design and development of the customized 3D model total-contact foot orthosis. (**a**) Foot impression from cast-taking simulation. (**b**) Foot contour sketching. (**c**) Final insole based on personal morphological characteristics.

To successfully import and numerically analyze the foot sole behavior within a customized full-contact insole under the same loading and boundary conditions, a new coefficient of friction of 0.5 was employed for foot–insole contact [34], and the same coefficient was used for plate–foot contact [27]. High-order 3D elements were established in the foot insole. From a high-order and semi-controlled discretization, 62,065 nodes and 34,816 elements were obtained (Figure 7). Furthermore, TPU mechanical properties were assigned to the designed orthotic device. These values were taken from previous research and literature (Table 3) [49,50].

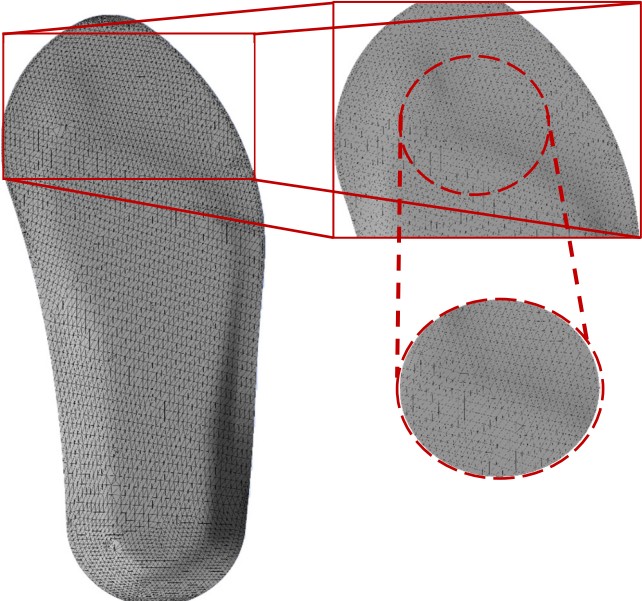

**Figure 7.** Foot orthosis discretization zoom-in views.

**Table 3.** Mechanical properties of TPU foot orthosis [49,50].

| Material | Young´s Modulus (MPa) | Poisson's Ratio |
|---|---|---|
| TPU | 11 | 0.45 |

### 2.4. Experimental Baropodometric Testing

A baropodometric study was performed to validate the reliability of the assumptions and the biofidelity of the model. The medical software FreeSTEP® v.1.4.01 was employed along with the professional equipment from Sensor Medica®, and the foot was evaluated statically (Figure 8a). The contact surface, percentage, and geometric values of the load applied on the foot were measured. Likewise, the study obtained results from a stabilometric analysis and a 3D scan of the foot. Once the calibration and adjustment of the equipment were completed, the pressure between the ground and the plantar surface of the foot was measured when the participant was standing barefoot on the platform, the distribution of the plantar pressure, the maximum plantar pressure, and the center of pressure were recorded. To more precisely determine the distributed load along the sole, the foot was divided into six parts automatically by the software (Figure 8b). Figure 8c shows the anthropometric measurements of both feet. Using the values and sections provided by the software, the experimental results obtained were compared with the model's predictions solved by numerical analysis.

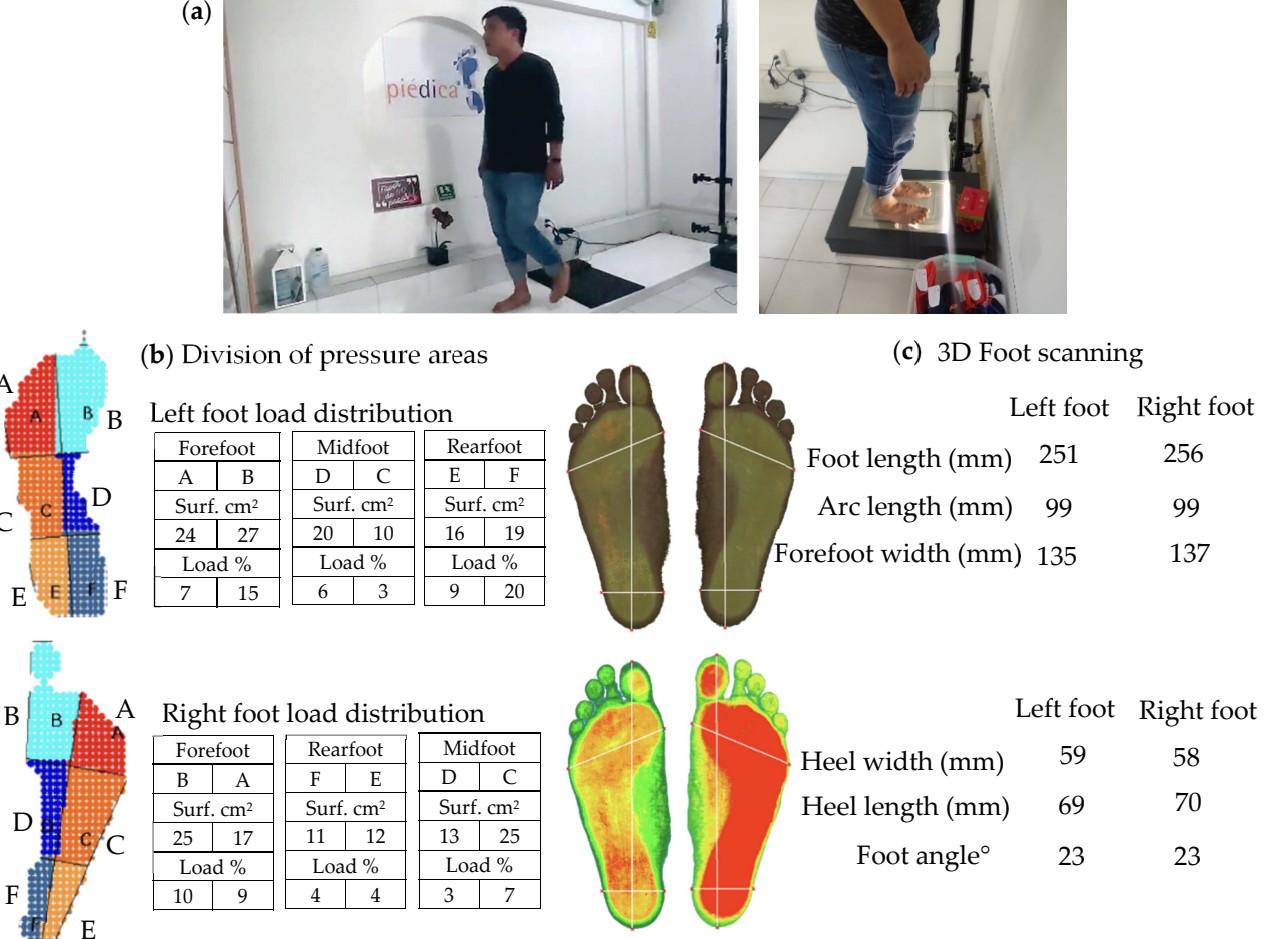

**Figure 8.** Baropodometric study. (**a**) Participant under experimental testing. (**b**) Sectioned regions of plantar pressure distribution. The abbreviation Surf in the tables refers to surface. (**c**) Three-dimensional anthropometric scanning.

## 3. Results

### 3.1. Footsketching Study Results

Several footprint sketches of the right foot of the participant were taken with different amounts of ink on the sole. These footprints were taken from a male young adult in his 30s. Using variations in the quantity of ink on the plantar surface allowed the observation of subtle changes in the plantar print that slightly changed the results. Despite slight variations in the percentages, they were very similar, resulting in the foot being classified as a normal foot type, as there was no tendency to fall into a Flatfoot–Normal or Normal–Cavus Foot classification. The results can be observed in Table 4 and Figure 9.

**Table 4.** Footprint results implementing the Hernández Corvo method.

| Footprint | Percentage % | Foot Type |
|-----------|--------------|-----------|
| a | 43.65 | Normal |
| b | 43.13 | Normal |
| c | 44.94 | Normal |
| d | 40.81 | Normal |

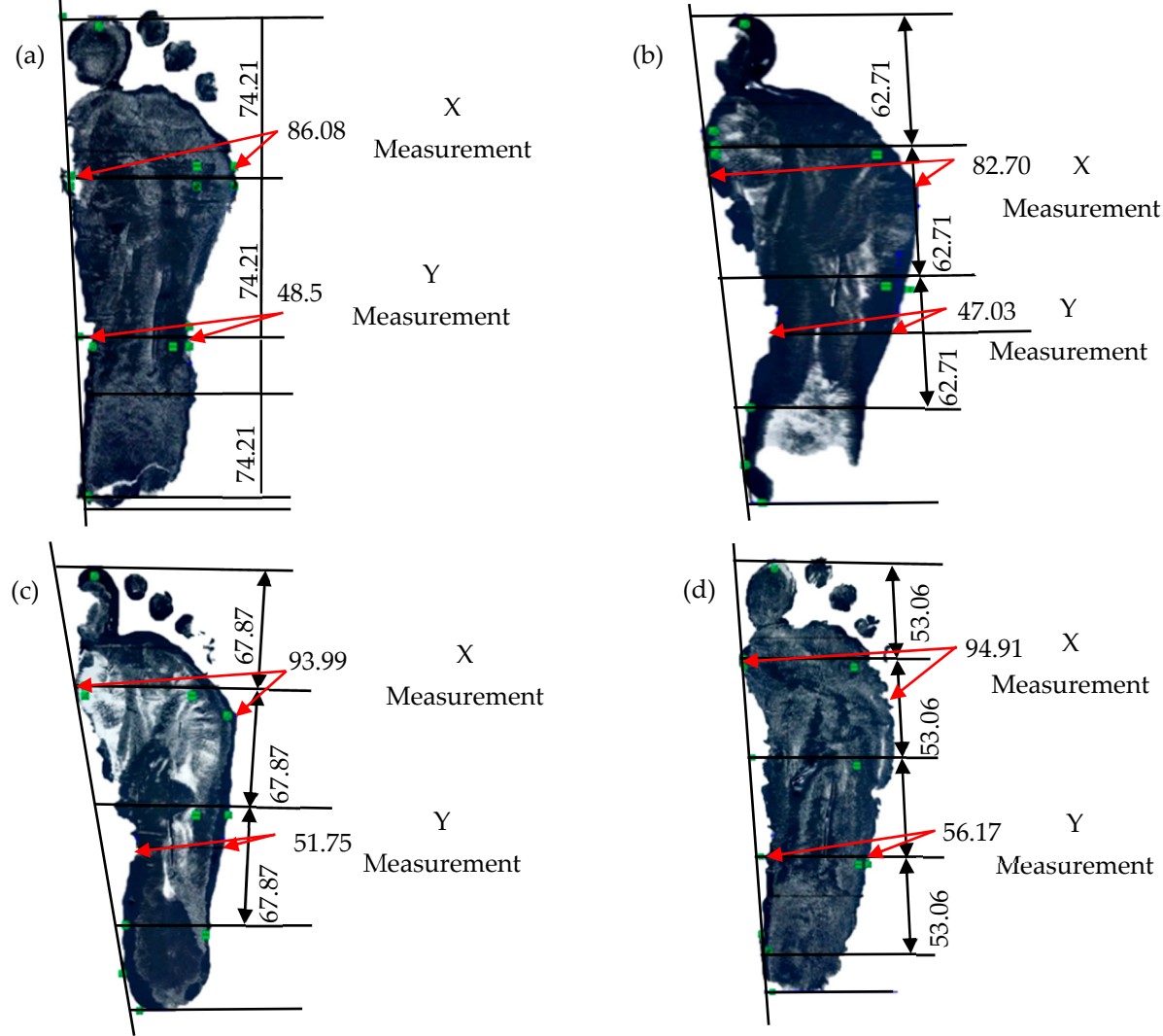

**Figure 9.** Footprint sketching results. (**a**) First trial. (**b**) Second trial. (**c**) Third trial. (**d**) Fourth trial.

### 3.2. First Case Study Results

Once the numerical analysis equations converged, results were obtained, mainly focusing on total deformation and von Mises stress due to representing a more precise behavior of the foot sole in both study cases. The visualization of the initial results corresponds to the less conservative model, corresponding to the mechanical properties proposed by Luboz (Figures 10 and 11), which has a lower Young's modulus value. The results obtained with the property defined by Wu are then shown (Figures 12 and 13).

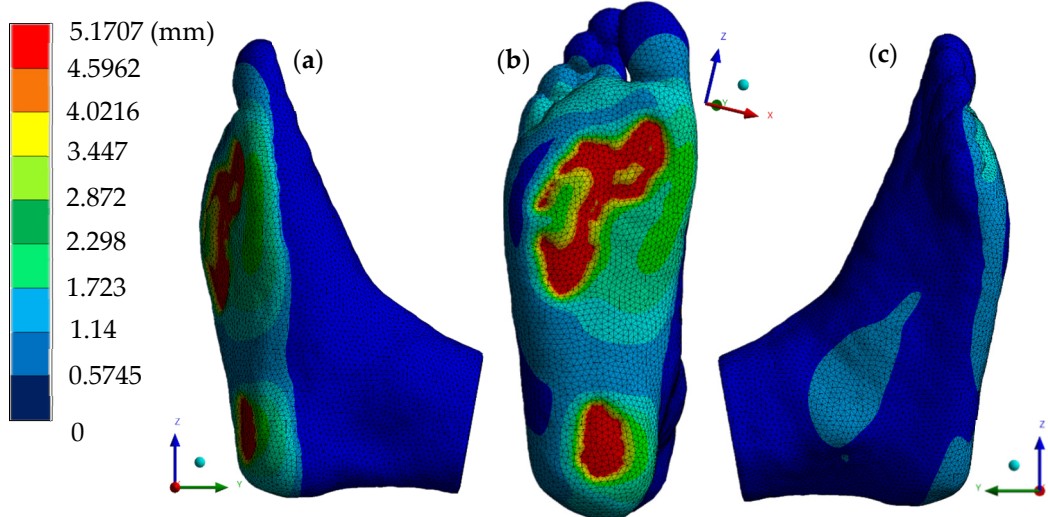

**Figure 10.** Total deformation (Luboz). (**a**) Left side view. (**b**) Plantar region. (**c**) Right side view.

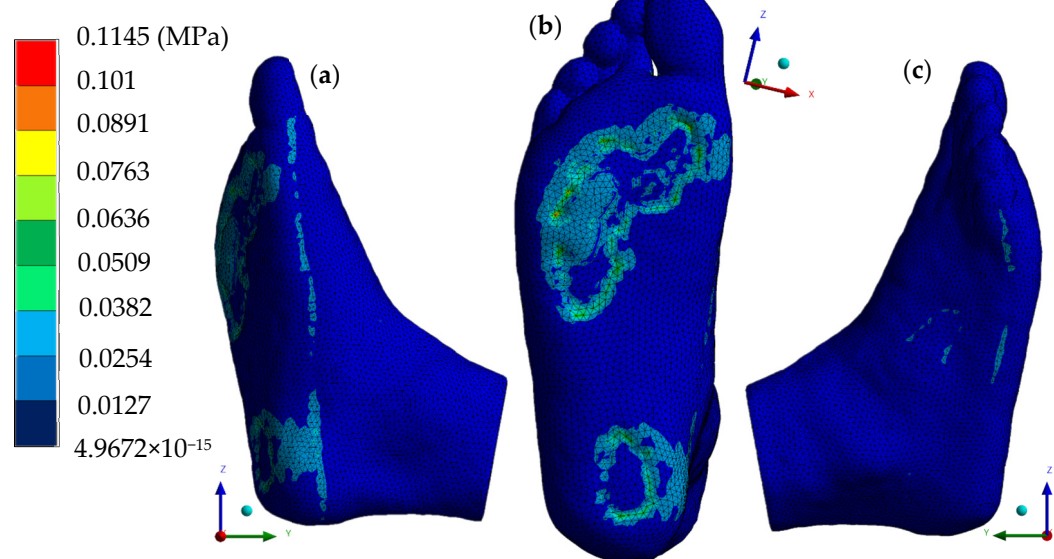

**Figure 11.** Von Mises stress (Luboz). (**a**) Left side view. (**b**) Plantar region. (**c**) Right side view.

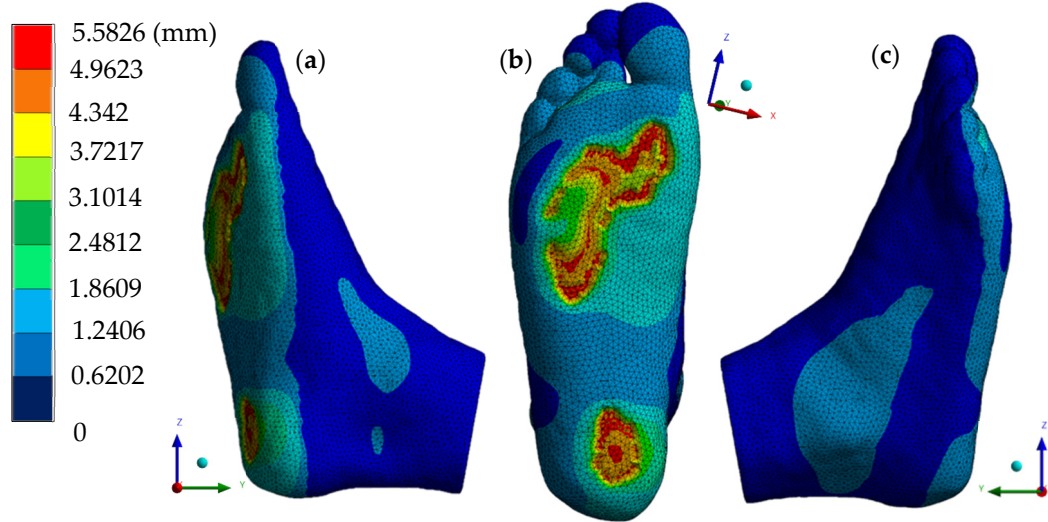

**Figure 12.** Total deformation (Wu). (**a**) Left side view. (**b**) Plantar region. (**c**) Right side view.

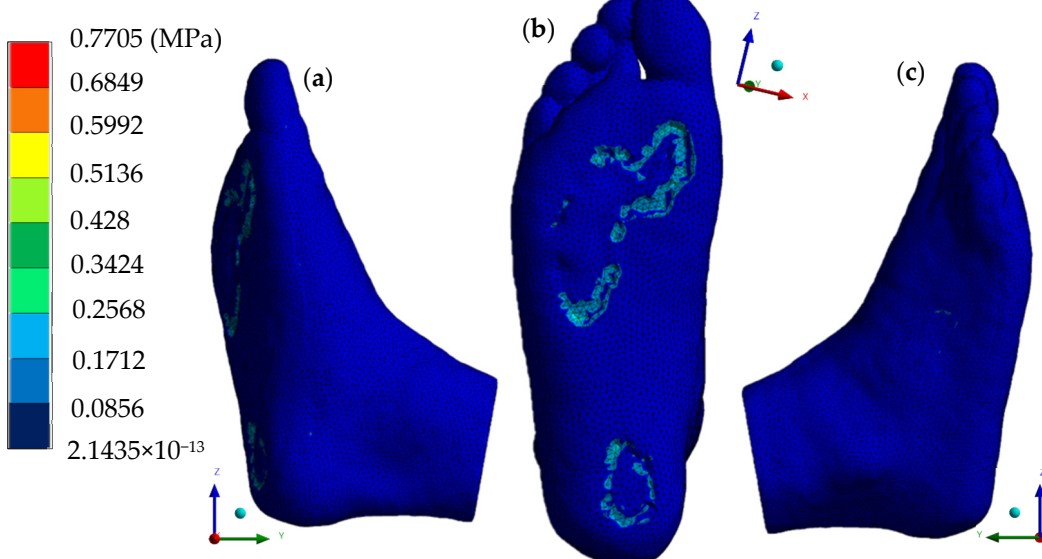

**Figure 13.** Von Mises stress (Wu). (**a**) Left side view. (**b**) Plantar region. (**c**) Right side view.

Results of Baropodometric Testing, Validation, and Comparison to First Case Study Results

A comparison of the plantar pressure points in the right foot sole between the experimental and numerical analysis proved the results to be extremely close for both mechanical properties for the encapsulating muscles, those proposed by Luboz and Wu. To compare and evaluate the stress concentration in these foot regions, von Mises stress theory values were considered; this theory is based on the difference in principal stresses. Therefore, this theory is optimal for the recreation of biological tissues since complex stress conditions and combinations of nominal and shear stresses are experienced in the plantar region under balanced standing. Furthermore, other researchers have widely used von Mises stress theory to measure and analyze the stress in the foot's plantar surface and soft tissues. The highest plantar pressure values obtained in the numerical analysis are around 0.050–0.063 MPa for the model with Luboz properties and 0.0856–0.1712 MPa for that with Wu properties. In comparison, the value registered in the baropodometric study yielded a 700 gr/cm$^2$ result, which is 0.0686 MPa (Figure 14). Thus, the numerical analysis predicted both highly trustworthy and precise results.

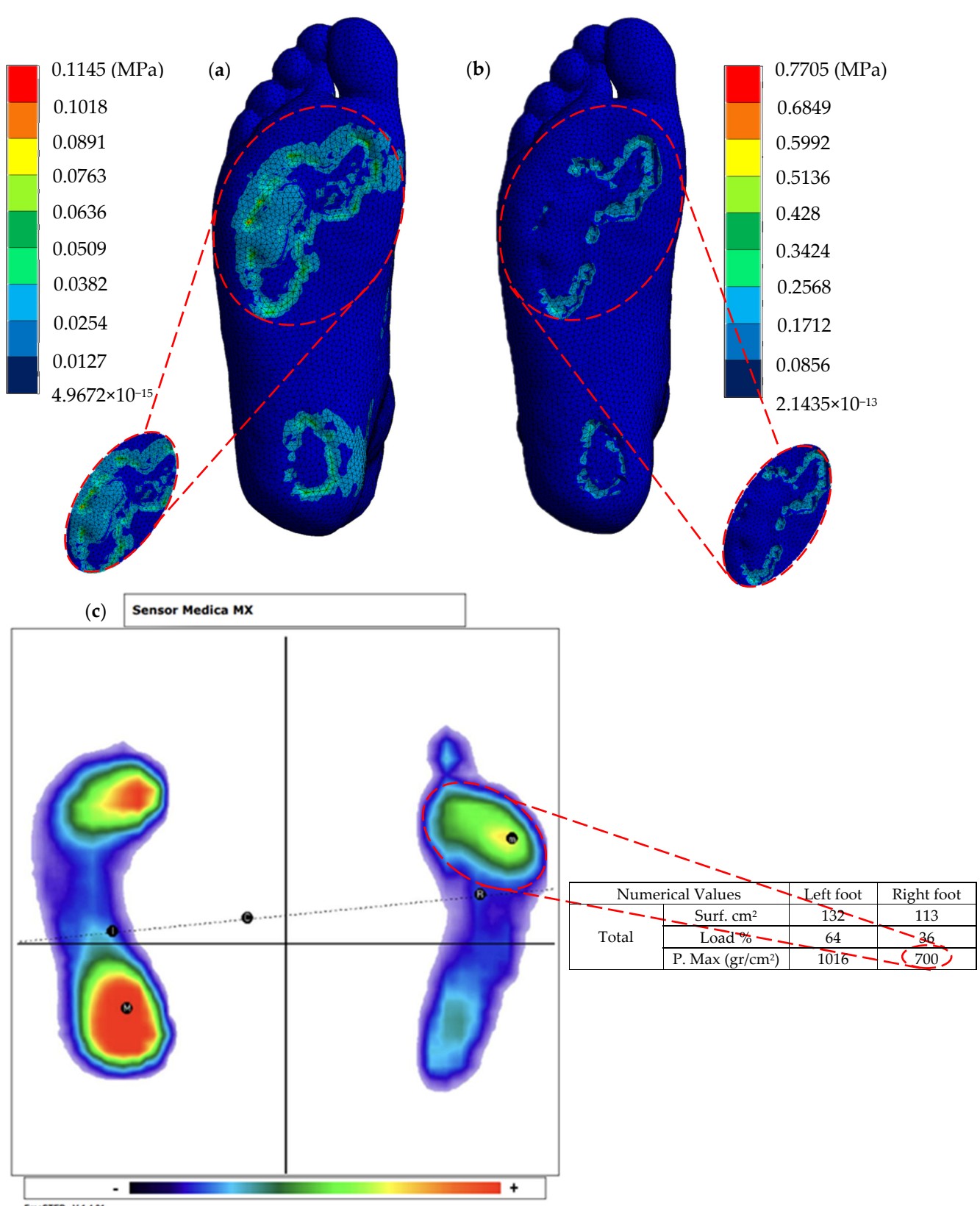

**Figure 14.** Comparison of numerical and experimental results. (**a**) Model with Luboz properties. (**b**) Model with Wu properties. (**c**) Baropodometric study. The abbreviation P. Max refers to maximum pressure.

### 3.3. Second Case Study Results

The results corresponding to the numerical evaluation employing the customized orthosis are presented for Luboz (Figures 15 and 16) and Wu (Figures 17 and 18), showing the cushioning effects redistributing pressure on the foot sole. All the maximum and minimum values obtained as results for both study cases can be found in Appendix A, Tables A1 and A2.

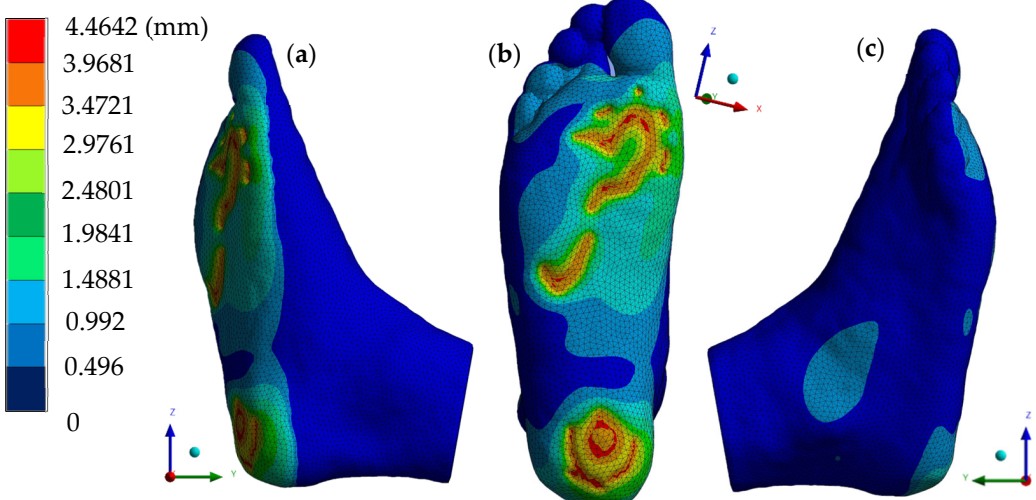

**Figure 15.** Total deformation with implementation of customized foot insole (Luboz). (**a**) Left side view. (**b**) Plantar region. (**c**) Right side view.

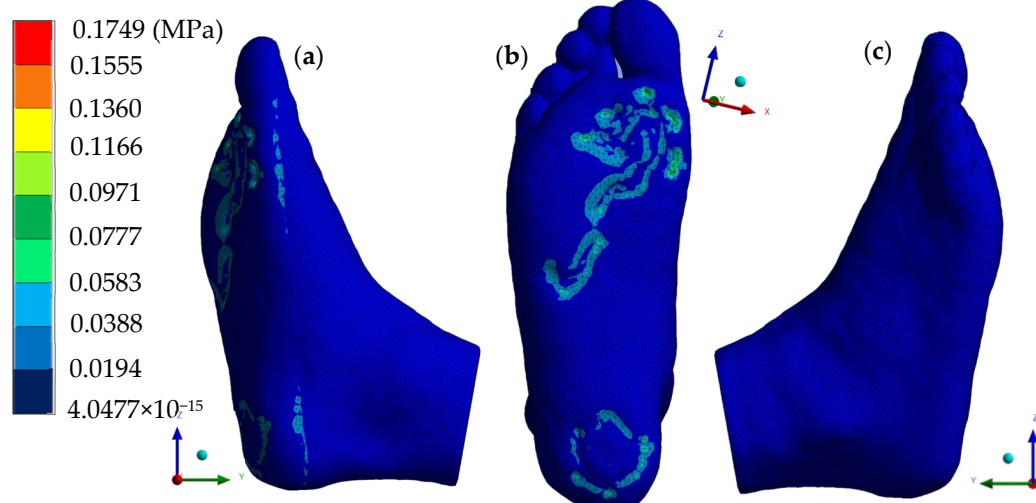

**Figure 16.** Von Mises stress with implementation of customized foot insole (Luboz). (**a**) Left side view. (**b**) Plantar region. (**c**) Right side view.

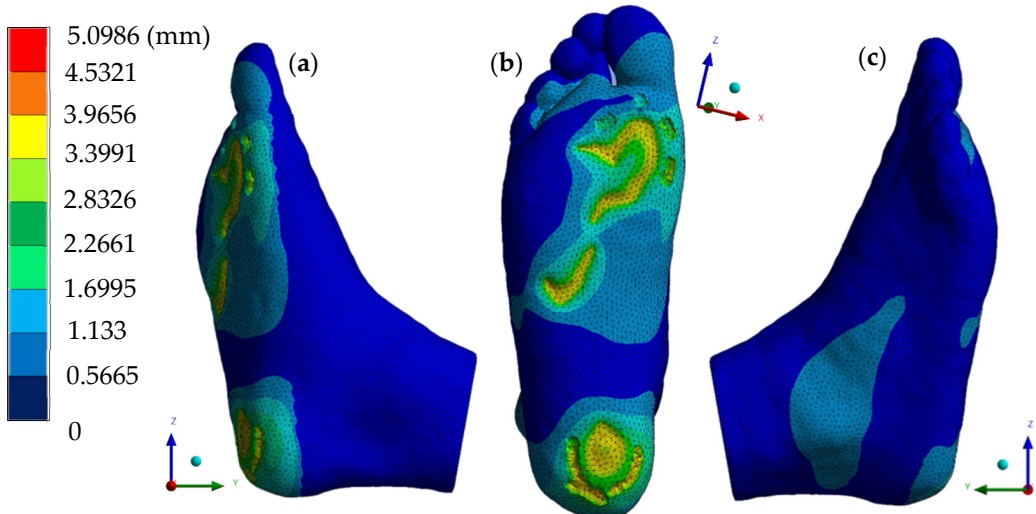

**Figure 17.** Total deformation with implementation of customized foot insole (Wu). (**a**) Left side view. (**b**) Plantar region. (**c**) Right side view.

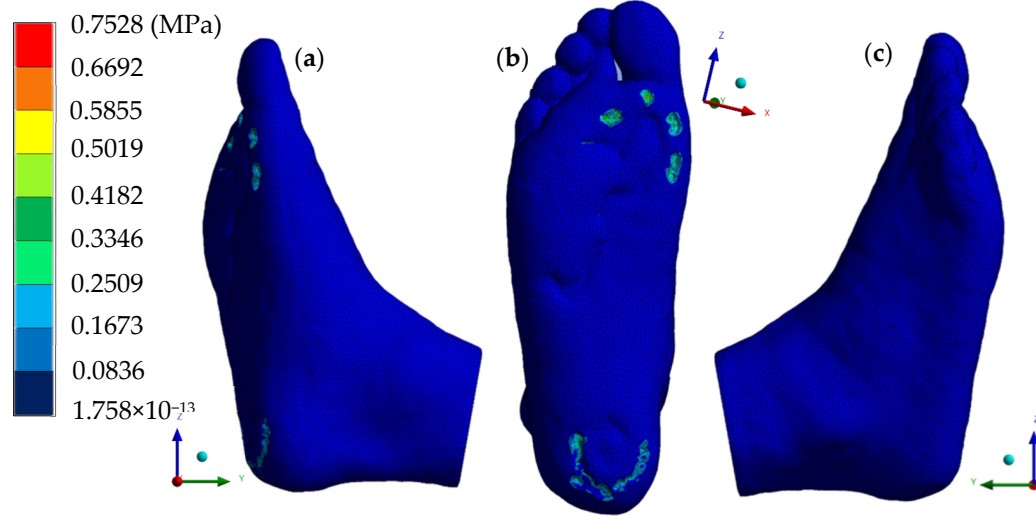

**Figure 18.** Von Mises stress with implementation of customized foot insole (Wu). (**a**) Left side view. (**b**) Plantar region. (**c**) Right side view.

## 4. Discussion

The human foot is among the most complex biological structures due to its remarkable functions, from locomotion to providing stability and support to the body, as it is the only human lower limb part having direct contact with the ground. Among these impressive functions, shock absorption is one of the most studied functionalities in medical and research fields, as it enables many functions, such as stabilization, body weight support, and surface adaptation.

A combination of experimental and numerical analyses allowed the acquisition of valuable data for analyzing the plantar foot region pressure points under normal standing conditions and providing a proper assessment in developing a personalized total-contact foot orthosis 3D model. This sophisticated methodology introduces segmentation procedures for reconstructing a three-dimensional foot model, representing an approach for feasibly reproducing different biological tissues. Despite representing foot muscles as an encapsulated element, the computational resource approach and intricateness set it apart as a highly elaborate three-dimensional biomodel construction.

In this research, mechanically recreating a static structural analysis in the standing condition predicted the appropriate behavior of the foot sole soft tissues, foot skin, and muscles. According to the medical literature, normal foot conditions present peak pressure points in the forefoot under the metatarsal heads, usually in the first or fifth, and the rearfoot in the heel while balanced standing is maintained. Numerical analysis showed a higher stress concentration in the forefoot region than in the rearfoot; mainly, the highest pressure was under the fifth metatarsal head. Thus, an equivalent representation in the plantar foot region for standard peak pressure values was obtained. In a validation of the finite element analysis, the pressure distributions were similar in the color scale depicted for both analyses. Thus, it was demonstrated that the participant tends to generate a higher pressure in the forefoot during balanced standing in normal conditions. Total elastic deformation results represent how the foot sole skin moves once it is in contact with the ground reaction force, having considerable values in regions with higher stress fields. Likewise, total elastic strain predicts the load tendency for the peak plantar pressure zones. The results for both models are in good agreement, adequately functioning when analyzed barefoot and within foot orthosis, employing the mechanical properties provided by Luboz and Wu. All the assumptions and considerations developed in the 3D modeling and finite element analysis for these foot soft tissues are considered precise. Since results are in good agreement and vary by very little, there is a standard error range between numerical and experimental testing.

The presented investigation stands out because it provides an innovative approach to analyzing the foot or any other biological tissue through finite element analysis since only soft tissues are considered. It is in contrast to most biological numerical analyses that require the reconstruction of bone tissue to analyze soft tissue shock absorption behavior. In addition, the detailed model developed and the fine discretization provide closer estimates as more differential partial equations converge into a solution.

Furthermore, both the material selection and geometry design of the customized 3D model insole were suitable because of the numerical prediction of lower peak pressure values and a uniform pressure redistribution along the foot sole, mainly reducing stress concentrations in most plantar regions where peak pressure points occurred when this anatomical position was maintained without the employment of any orthotic device. Foot sole regions presented a minor pressure increment, resulting in foot–insole contact. The initial numerical analysis utilizing the finite element method provides an appropriate assessment for the geometric design of a 3D model customized insole evaluation, firstly relying on analyzing the foot by itself (barefoot) and then considering both morphological and anthropometrical aspects to comprehend where there is a higher likelihood of peak pressure points occurring. The plantar orthosis material also has a significant role in the correct performance, commonly based on controlling foot functions or providing cushioning effects. TPU was ideal for relieving peak pressure points on tender spots, giving additional support, enhancing stability, and adding an extra layer to the plantar region.

Numerically analyzing pressure points in a foot, apparently under normal conditions, promotes a more thorough comprehension of real-life behavior under the simulated anatomical position. Deepening current knowledge about this subject could better implement numerical approaches for pathological foot analysis, giving proper medical evaluation towards rehabilitation. In addition to experimental validation, the present study closely aligns with extant investigations in foot finite element analysis modeling. Notably, whereas prior published studies predominantly focused on incorporating bony elements within their models, this work presents a unique aspect by exclusively considering the foot's soft tissues. This methodological distinction deviates from established approaches and thus represents a novel contribution to the existing body of literature. Numerical simulations' stress distribution results are in solid concordance with previous finite element analyses when evaluating barefoot balanced standing, reinforcing the observations reported regarding peak pressure values and the distribution of stress patterns in the plantar region [51–53], specifically when utilizing the muscles' mechanical properties provided by Luboz. At

the same time, results obtained in the plantar surface, including the customized insole, correlate highly with other research papers that optimize insole design through numerical analyses [54–58]. Moreover, the plantar peak pressure values for the utilization of the personalized orthotic device are in substantial consensus with recent literature in insole construction combining additive manufacturing materials and traditional materials [49,59]. Nonetheless, the specific reconstructed model features differ from most foot finite element models in the literature; the results are in accordance due to two main reasons: utilizing a similar methodology to analyze balanced standing and the reconstruction of healthy foot models in similar patient populations in healthy conditions without foot pathology issues.

Despite this study demonstrating feasible results to be established as a solid methodology for numerical analyses in biological tissues, it is relevant to point out certain limitations of this research. While it is not a significant issue in employing this methodology, it is relevant to account for proper computational equipment to develop the numerical analyses quickly. Another issue that may compromise the reproducibility of this paper relies on the need for medical data to assign displacement magnitude, taking into account that defining the displacement value has a relationship with body weight but does not have a direct conversion. A fundamental limitation of this study is the use of data from a single, young, and healthy subject. This consideration restricts the generalizability of our findings to broader populations with diverse health conditions. While the proposed approach demonstrates promise in this specific case, further research is necessary to validate its efficacy and applicability in individuals with various health profiles. Future studies should include more extensive and diverse participant pools, incorporating individuals with various medical conditions and demographic characteristics.

## 5. Conclusions

Finite element analyses have been established as a powerful tool for evaluating biological tissues, providing valuable insights into understanding their complex behavior. This numerical engineering technique has the ambition to generate an even higher impact on the medical field. Notably, in the presented research, a proper assessment for creating a refined personalized 3D model orthotic device was employed through a numerical evaluation. Nonetheless, to use the described technique, it is mandatory to have strong mechanical knowledge and a high degree of expertise in segmentation and numerical analysis software in addition to having powerful computational equipment so that the methodology will not be time-consuming since biomodel refinement (biofidelity) and finite element analysis are directly related to computational features.

Numerical analysis can be applied to numerous approaches that, along with medical supervision, can trigger more sophisticated techniques when evaluating the outstanding but always complicated human body. It is relevant to mention that numerical analysis cannot replace experimental testing but results in an advantageous methodology complementing existing medical procedures predicting when the body may be susceptible to injuries and taking action when they occur. Furthermore, the interdisciplinary approaches from the union of medicine and engineering, biomechanics, and biomedicine, to mention a few, have facilitated the enhancement of current prostheses, orthoses, pre-surgical assistance and planning, and rehabilitation therapies. Moreover, the accelerated growth of additive manufacturing technologies has enabled new findings regarding new materials in assistive devices, with particular advantages such as easy access, affordability, and time efficiency.

Considering everything, the finite element analysis employed in this research can obtain estimations close to reality, validating engineering and mathematical methods as a reliable complementary tool regarding complex clinical assessment. Thus, the methods applied in the present work can change how traditional customization procedures in the medical field are currently carried out, with the concrete aim of creating personalized prosthetic and orthotic devices since the results obtained substantiate the utilization of numerical analysis in biological tissues, accurately predicting the behavior of these tissues

under concrete circumstances. These methods provide an alternative to standard clinical procedures that are often time-consuming and expensive.

**Author Contributions:** Conceptualization, J.A.S.-P., G.U.-S. and B.R.-Á.; methodology, J.A.S.-P., G.U.-S., B.R.-Á. and G.M.U.-C.; validation, J.A.S.-P., G.U.-S., B.R.-Á., G.M.U.-C. and S.C.-L.; formal analysis, J.A.S.-P., G.U.-S., B.R.-Á. and D.E.C.-L.; investigation, J.A.S.-P., G.U.-S., B.R.-Á. and G.M.U.-C.; resources, J.A.S.-P., G.U.-S., B.R.-Á. and J.R.G.-I.; writing—original draft preparation, J.A.S.-P., G.U.-S., B.R.-Á. and G.M.-A.; writing—review and editing, J.A.S.-P., G.U.-S., B.R.-Á. and A.U.-L.; visualization, J.A.S.-P., G.U.-S., B.R.-Á. and A.U.-L.; supervision, J.A.S.-P., G.U.-S. and B.R.-Á.; project administration, J.A.S.-P., G.U.-S., B.R.-Á. and G.M.U.-C. All authors have read and agreed to the published version of the manuscript.

**Funding:** This research received no external funding.

**Institutional Review Board Statement:** The study was conducted in accordance with the Declaration of Helsinki and approved by the Ethics Committee of Biomechanics Group, Instituto Politécnico Nacional, Escuela Superior de Ingeniería Mecánica y Eléctrica, Sección de Estudios de Posgrado e Investigación, Unidad Profesional Adolfo López Mateos, 14 June 2023.

**Informed Consent Statement:** Informed consent was obtained from the participant involved in the study. Written informed consent has been obtained from the participant to publish this paper.

**Data Availability Statement:** All data generated or analyzed during this study are included within the article.

**Acknowledgments:** The authors gratefully acknowledge the Instituto Politécnico Nacional, Consejo Nacional de Humanidades Ciencias y Tecnologías, and Francisco Carrasco Hernández for supporting this research.

**Conflicts of Interest:** The authors declare no conflicts of interest.

## Appendix A

**Table A1.** First case study summary of numerical evaluation results.

| Type of Analysis | Luboz Properties | | Wu Properties | |
|---|---|---|---|---|
| | **Maximum** | **Minimum** | **Maximum** | **Minimum** |
| Total deformation (mm) | 5.1707 | 0 | 5.5826 | 0 |
| Deformation X axis (mm) | 2.4654 | −1.403 | 1.7405 | −1.2755 |
| Deformation Y axis (mm) | 5.1707 | −0.7618 | 5.5737 | −0.5247 |
| Deformation Z axis (mm) | 1.8151 | −1.6289 | 1.5674 | −1.5781 |
| Total elastic strain (mm/mm) | 0.6602 | $4.5741 \times 10^{-16}$ | 1.6765 | $4.4235 \times 10^{-16}$ |
| Elastic strain X axis (mm/mm) | 0.3623 | −0.3095 | 0.9604 | −0.6028 |
| Elastic strain Y axis (mm/mm) | 0.5329 | −0.5899 | 0.7187 | −1.4239 |
| Elastic strain Z axis (mm/mm) | 0.3541 | −0.2437 | 0.6552 | −0.5808 |
| Nominal stress X axis (MPa) | 0.1058 | −0.1357 | 0.3503 | −1.6429 |
| Nominal stress Y axis (MPa) | 0.1201 | −0.179 | 0.3885 | −1.8739 |
| Nominal stress Z axis (MPa) | 0.1014 | −0.1526 | 0.2811 | −1.6988 |
| Shear stress XY plane (MPa) | 0.0430 | −0.038 | 0.2995 | −0.2955 |
| Shear stress YZ plane (MPa) | 0.0357 | −0.0373 | 0.1983 | −0.2349 |
| Shear stress XZ plane (MPa) | 0.0217 | −0.0203 | 0.1292 | −0.1208 |
| von Mises stress (MPa) | 0.1145 | $4.9672 \times 10^{-15}$ | 0.7705 | $2.1435 \times 10^{-13}$ |
| Maximum principal stress (MPa) | 0.1222 | −0.1313 | 0.4689 | −1.5912 |
| Minimum principal stress (MPa) | 0.1010 | −0.19 | 0.1887 | −1.9227 |

**Table A2.** Second case study summary of numerical evaluation results.

| Type of Analysis | Luboz Properties | | Wu Properties | |
|---|---|---|---|---|
| | **Maximum** | **Minimum** | **Maximum** | **Minimum** |
| Total deformation (mm) | 4.4642 | 0 | 5.0986 | 0 |
| Deformation X axis (mm) | 1.9254 | −1.1515 | 1.9025 | −1.2693 |
| Deformation Y axis (mm) | 4.4627 | −0.5956 | 5.0832 | −0.4367 |
| Deformation Z axis (mm) | 1.29 | −1.3982 | 1.5646 | −1.8064 |
| Total elastic strain (mm/mm) | 0.9865 | $7.3806 \times 10^{-14}$ | 2.6486 | $3.5677 \times 10^{-16}$ |
| Elastic strain X axis (mm/mm) | 0.5086 | −0.2228 | 1.034 | −0.6846 |
| Elastic strain Y axis (mm/mm) | 0.4684 | −0.8719 | 0.8662 | −2.0931 |
| Elastic strain Z axis (mm/mm) | 0.3331 | −0.3519 | 1.1336 | −0.6302 |
| Nominal stress X axis (MPa) | 0.0759 | −0.2348 | 0.4053 | −1.5162 |
| Nominal stress Y axis (MPa) | 0.0975 | −0.4690 | 0.3665 | −2.2535 |
| Nominal stress Z axis (MPa) | 0.0858 | −0.3110 | 0.3382 | −1.8223 |
| Shear stress XY plane (MPa) | 0.0585 | −0.0548 | 0.4901 | −0.3188 |
| Shear stress YZ plane (MPa) | 0.0529 | −0.0417 | 0.3888 | −0.3655 |
| Shear stress XZ plane (MPa) | 0.0240 | −0.0173 | 0.1382 | −0.1987 |
| von Mises stress (MPa) | 0.1749 | $4.0477 \times 10^{-15}$ | 0.7528 | $1.758 \times 10^{-13}$ |
| Maximum principal stress (MPa) | 0.1262 | −0.2298 | 0.5284 | −1.4975 |
| Minimum principal stress (MPa) | 0.0646 | −0.4761 | 0.3214 | −2.4995 |

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
