# Peer review of "Biomechanical Evaluation of Plantar Pressure Distribution towards a Customized 3D Orthotic Device: A Methodological Case Study through a Finite Element Analysis Approach"

_applsci, doi:10.3390/app14041650_

Round 1
Reviewer 1 Report (Previous Reviewer 1)
Comments and Suggestions for Authors
The authors improved the paper, but I still have two concerns, and suggest small changes:
1. Title – this is a methodological study, showing the application of the modelling based on the image data and data from only one, healthy subject. This is not reflect by the title, which, in my opinion, should be changed. I suggest addition of the second part, for example: A methodological study, or Presentation of a method, etc.
2. Limitations – one of the most important limitations is using the data of one, young and healthy subject. This should be stressed in the Limitation section of the paper, and the need to perform similar studies on at least few patients with various health issues to verify the applicability of the proposed approach.
Author Response
Please see the attachment

Reviewer 2 Report (Previous Reviewer 3)
Comments and Suggestions for Authors
The authors have taken into account or responded to my comments. I believe that the article is now ready for publication.
Author Response
Please see the attachment

Reviewer 3 Report (Previous Reviewer 4)
Comments and Suggestions for Authors
This version is much better than the previous version and still require some changes. More specifically, the authors somehow need to discuss the merit of their research compared with previous literature. Compare and contrast the results they have obtained and how they are similar or different.
1. "a Computed Tomography (CT) scan was used on 30-year- old Mexican young adult"
Provide the type of CT scan machine used: Model, resolution.
What is the distance between slices?
2. "30-year-old Mexican young adult in apparently healthy condition ....foot in normal conditions"
It will be good to add weight and height, or BMI information.
Also need to provide some information of the foot, example length and width.
3. Planes view during the segmentation process
It is useful to give a location of the sections (either in percentage or mm)
4. Provide some info about the CAD model, example number of nodes before and after remeshing.
5. The discussion is very weak. Need to discuss how your research agrees or disagrees with already published papers of foot FEA model and FEA model of foot and insoles.
Author Response
Please see the attachment

Reviewer 4 Report (Previous Reviewer 5)
Comments and Suggestions for Authors
The revise paper has improved and well illustrated the topic.
Author Response
Please see the attachment

This manuscript is a resubmission of an earlier submission. The following is a list of the peer review reports and author responses from that submission.
Round 1
Reviewer 1 Report
Comments and Suggestions for Authors
This paper deals with an important issue: customized insoles. It applies the modelling method with finite elements and pressure distribution measurements. It is an interesting study, which merits publication, but some issues should be addressed by the authors.
1. This is a methodology description with a young healthy man as a subject. Healthy subjects rarely use insoles, and this methodology should be also verified on patients with foot deformities. The title should reflect this fact.
2. The serious limitation of the study is the static condition of measuring the pressure distribution. During dynamic tasks such as walking the active muscles can change the shape of the foot, and a flat foot while standing is normal while walking. Humans sometimes stand a lot, but we mainly walk, and if the insole is designed based on static data there is a chance that its action while walking would be nefarious.
3. I do not agree with the aim formulated as: "This research aims to enhance the knowledge and perceptions of the biomechanical behavior of this lower extremity and provide a real impact in assessing the design of personalized plantar orthoses". This is a modelling study based on the external shape of the foot and pressure distribution. Such an approach does not add anything to our knowledge of the biomechanics of the foot, musculoskeletal actions, or internal forces. The aim should reflect what was really done.
4. The limitations of this study should be described and discussed, and their possible influence on the conclusion clearly presented.
Reviewer 2 Report
Comments and Suggestions for Authors
Thank you for the opportunity to review the manuscript titled "Biomechanical evaluation of plantar pressure distribution towards a customized 3D orthotic device: A Finite Element Analysis approach." The manuscript describes a study evaluating the precision of pressure-point estimation using biomodelling and Finite Element Analysis of a normal foot during standing. The authors claim to have developed an innovative approach to analyzing the foot that can change customization procedures for personalized prosthetic and orthotic devices.
1. Unfortunately, I am not convinced that the described approach is as revolutionary as claimed. It requires the use of CT, which is difficult to access and expensive to use, powerful computational equipment, which is often not readily available, specialized commercial software, and a high degree of expertise in using the software. It is unclear how this helps solve the "common problems for Biomechanics" mentioned in the Introduction, i.e., the need for professional equipment to perform studies and the requirement to perform in-vivo tests. This should be additionally explained.
2. It is also unclear why the "Second case study" was performed if the insole was not manufactured and tested in the baropodometric experiment to confirm the results of the simulation. There is nothing inherently novel about modelling a hyper-customized insole and analyzing it via FEM, and the results of FEA appear to have only been validated for the "First case study". The last sentence of the Conclusion should be substantiated ("Thus, methods applied in the present work can change how traditional customization procedures in the medical field are currently carried out, with the concrete aim of creating personalized prosthetic and orthotic devices.").
3. The insole was designed to have "a 3 mm thickness" (Line 361). Was this thickness uniform across the entire insole? Namely, if the surface of the insole interfacing with the foot closely followed the morphology of the foot in real life, should the part in contact with footwear not follow the shape of the shoe to provide effective cushioning when standing?
4. The output of the Baropodometric study in Figure 14 appears to suggest that the participant's weight was very unequally distributed between the right and left foot. Although considerably higher pressures were recorded at the left foot, only data for the right foot was analyzed in the study. This requires an explanation.
5. The manuscript should be restructured and thoroughly rewritten, as it is at times very difficult to read and comprehend. At present, some results are included in the "Materials and Methods" section (e.g., Table 2, Figure 1), and some materials and methods, as well as results are described in sections following "Materials and Methods" (i.e., under "Numerical analysis of the foot biomodel" and "Experimental baropodometric testing"). The authors should consider thoroughly revising the structure of the manuscript in line with academic writing guidelines.
5.1. The Materials and Methods section could be improved if it were divided into the basic elements: Study design, Participant, Materials and setting, Data analysis, and Ethical approval. The description of the participant should be moved under the heading "Participant" in the Materials and Methods section. It is also advised to avoid referring to participants as "study subjects" (Line 178); the term "participant" is preferred.
5.2. If my understanding is correct, "Footprint Sketching" was only performed to determine whether the participant's foot was of a "Normal" type, as required for the study (i.e., people with non-normal foot types would not be eligible to participate in the study). At present, the section describing this procedure is unnecessarily long and detailed, considering that the results were only used to confirm the participant's eligibility.
6. Both studies mentioned in Line 280 ("... foot biomechanical models by Perrier and Nafiseh [28,29].") used and cited earlier sources for the Young's Modulus and Poisson's Ratio of foot muscles; please refer to the original studies: Perrier -> Luboz et al. (2015); "Nafiseh" is the first name of the author of Ahanchian et al. (2017) -> Wu (2007).
7. The meaning of symbols used in Figures 4 and 6 (U, Rot) should be explained in the figure caption.
Comments on the Quality of English LanguageAWKWARDLY WORDED SENTENCES
Several sentences are very awkwardly worded and should be rephrased to improve understandability, including the following:
- Lines 18-19: "Experimental testing is currently employed to study static foot conditions, which are invasive and noninvasive techniques."
- Line 25: "... the first was developed to evaluate foot behavior deformation ..."
- Line 29: "... the first study case demonstrated successfully predicted ..."
- Lines 30-31: "Employing a customized insole resulted in exceptionally beneficial in its primary function ..."
- Lines 56-58: "These tests are remarkable efforts to understand the morphological foot mechanism to distribute the load within its unique adaptation to different ground geometries."
- Line 68: "... generated through the application of the medical branch of imaging ..."
- Lines 78-80: "This research aims to enhance the knowledge and perceptions of the biomechanical behavior of this lower extremity and provide a real impact in assessing the design of personalized plantar orthoses."
- Lines 95-96: "Usually, the working procedure for this methodology is sketching techniques using the photopodogram or the pedigraph."
- Lines 97-98: "... photopodogram technique uses ink or paint to paint the foot and step on thermographic paper over a flat surface."
- Lines 170-172: "... a Mexican adult in healthy conditions of 30 years old with a foot and complexion in normal conditions, with a tendency to practice sports."
- Lines 223-225: "... to close and further refine the model to obtain a solid representation, closing cavities or empty improper spaces from the segmentation process."
- Lines 226-228: "The edges were refined by smoothing them, assembling the two solid elements, and re-meshing was added to the model to optimize its handling when numerically analyzing it (Figure 3b)."
- Lines 249-250: "... the foot is analyzed as a structure with a load in compression towards the plantar surface."
- Lines 256-257: "As shown in Figure 4, the representation of a free-body diagram for the loading and boundary conditions."
- Lines 280-282: "The choice of 2 mechanical properties for the muscle relied on developing an analysis with a partially conservative posture."
- Line 282: "... the value of the mechanical property of the plate ..."
- Lines 312-314: "The application of the external agent is assumed as the plate performing a vertical displacement of 5 mm penetrating the plantar surface of the model to produce vertical loads since the weight of the person's foot analyzed is 80 kg."
- Lines 314-315: "According to experimental bases, there is a relationship when exerting a force of 400 N in each foot."
- Lines 317-320: "In addition, there is evidence that applying a loading condition within a displacement acting as a pressure generates estimations closer to the natural behavior of the biomechanical characteristics of the plantar surface [29-33]."
- Line 347: "... resulting convenient to handle with proper 3D printing hardware."
- Lines 350-351: "For the design of the foot orthosis, plenty of methodologies were reviewed to design the most accurate and closest to the specific foot morphology of the subject of study."
- Lines 523-525: "... such theory is based on the difference of principal stresses as being the most reasonable and ideal. Since complex stress conditions, combinations of nominal and shear stresses are experienced in the plantar region under balanced standing."
- Lines 531-532: "Thus, the high trustworthiness of the results obtained from the numerical analysis was accomplished, and so was their accuracy."
- Lines 650-653: "Among these impressive functions, shock absorption is amongst the most interesting in medical and research fields. Developing plenty of functions such as stabilization, body weight support, and surface adaptation are a few of them."
- Lines 657-659: "The methodology presented involves innovative procedures to reconstruct a three-dimensional foot model, making this sophisticated approach feasible to reproduce different types of biological tissues."
- Lines 666-669: "Numerical analysis showed a higher stress concentration in the forefoot region, particularly under the fifth metatarsal head, than in the rearfoot, obtaining an equivalent representation in the plantar foot region within standard peak values."
- Lines 675-677: "Both models have different mechanical properties for the muscles to work appropriately, having similar behavior when analyzed barefoot and within foot orthosis; Perrier and Nafiseh."
- Lines 701-704: "Numerically analyzing pressure points in a foot apparently under normal conditions promotes a more thorough comprehension of how it should behave under regular conditions. By being knowledgeable about this subject, a pathological foot analysis could be better implemented, giving proper medical evaluation towards rehabilitation."
- Lines 706-707: "Numerically evaluating biological tissue, Finite Element Analyses have been demonstrated to have immense value when considering biological tissue."
REDUNDANT SENTENCES
Some sentences appear to be unnecessary - please consider removing them:
- Lines 108-110: "Once the methodology literature was reviewed to develop this technique, the previously described method obtained the plantar impression using the photopodogram technique."
- Lines 162-164: "Prior knowledge of medical imaging and its operation protocol is fundamental since 162 this is the basis for developing computer programs capable of processing all the medical 163 information stored in the DICOM format in images."
- Lines 195-196: "The characteristics of foot muscles need to be studied more due to their complexity."
MINOR PROOFREADING ISSUES
- Line 27 (and multiple other instances): "3D fully contact foot orthosis" -> "total contact foot orthosis". Additionally, what is the purpose of calling it a "3D" orthosis - are not all orthoses three-dimensional? Should this be "3D-printed" orthosis or a "3D model" of the orthosis?
- Line 43 (and multiple other instances): "plenty of" -> "many"/"several"
- Line 52: "... affect the foot organs internally ..." -> "... affect internal foot tissues ..."
- Line 63: "... typically, to obtain better results; it is required ..." -> "... typically, to obtain better results, it is required ..."
- Line 91: "... static methodologies exist to obtain such footprint ..." -> "... static methods exist to obtain such a footprint ..."
- Line 92: "these methodologies" -> "these methods"
- Line 114-115: "... classified as a solid foot under normal conditions ..." -> "... classified as a Normal foot type ..." (as per the classification in Table 1)
- Line 170: "... a computed tomography, CT, scan was used ..." -> "... a Computed Tomography (CT) scan was used ..."
- Line 177: "A total of 1071 slices in all anatomical planes was obtained." -> "A total of 1071 slices in all anatomical planes were obtained."
- Line 186: "... interested soft tissue areas ..." -> "... soft tissue areas of interest ..."
- Lines 182-192: "Develop" -> "Development"; "Determine" -> "Determination"; "Export" -> "Exportation"; "solidify" -> "solidification"; "apply" -> "application"
- Line 189: "On Materialise 3-Matic" -> "In Materialise 3-Matic"
- Line 277: "Precisely, for this study, ..." -> "For this study, ..."
- Line 279: "... homogeneous, and isotropic behavior. Hence, taking the values ..." -> "... homogeneous, and isotropic behavior, taking the values ..."
- Line 288: "... for saving ..." -> "... to save ..."
- Line 308: "... the forefoot, and around are embedded." - What does "and around" mean here? Please rephrase
- Lines 339 and 343: The full name of TPU should be spelled out when it is first mentioned; i.e., Thermoplastic polyurethane (TPU); afterwards, only the abbreviation should be used.
- Lines 339-340: "the contact foot sole" - Why "contact"?
- Lines 341-342: "... the insole cushioning effects absorb most ground reaction forces and visualize biomechanical results." - What does "visualize biomechanical results" mean here? Please rephrase
- Line 344: "several recent research" -> "several recent studies"
- Line 346: "fused filament fabrication, FFF," -> "Fused Filament Fabrication (FFF)"
- Line 348: TPU has "ideal mechanical properties for stress redistribution, compression strength, and pain relief;" - What does "compression strength" mean here? Please rephrase
- Line 349: What does "sustainably advantageous" mean? Please rephrase
- Line 353: "positive foot impression" - Is "positive" correct here?
- Line 359: "... certain regions to be smoothed and adjust the orthosis according to foot morphology ..." -> "... certain regions to be smoothed and the orthosis adjusted to foot morphology ..."
- Line 363: "... unimportant for materials made from TPU [48]." - This is unclear, TPU itself is a material
- Line 386 and 692: "foot-foot insole" -> "foot-insole"
- Line 564: "Subsequently following this order, results corresponding ..." -> "Results corresponding ..."
- Line 568: "observed" -> "found"
- Line 650: "limb" - The foot is part of the lower limb, it is not a limb itself
- Line 654: "A combination of experimental and numerical analysis ..." -> "A combination of experimental and numerical analyses ..."
- Line 673: "... foot sole skin moves along ..." -> "... foot sole skin moves ..."
- Line 688: "... both material selection and geometry design of the customized 3D insole were suitable due to numerically predicting lower peak pressure values ..." - Is "due to" correct here?
- Line 697: "3D Plantar orthosis material" -> "Plantar orthosis material"
Reviewer 3 Report
Comments and Suggestions for Authors
I consider the article received for review to be very relevant and essential for future research and development of medical diagnostics.
I have a few comments that will help the authors to improve their work.
1. were the footprints taken from only one man? Were the results obtained and their compatibility between the different methods used by the authors checked on a larger number of people?
2 In order to make the text easier to understand, I recommend:
(a) under figures 4 and 6, explain the abbreviations used (U, Rot)
b) When using the abbreviation TPU for the first time, the full name of the material should be used.
c) In line 343 "Thermoplastic polyurethane, TPU, employment as the insole material", if TPU is the abbreviation of Thermoplastic polyurethane it should be put in brackets.
d) In Figure 9 b) in the table, concerning Midfoot left foot load distribution, C should be used instead of R.
e) In Figure 9 b) the letters indicating the parts of the foot should be clearer. In the drawing of the right foot, it looks as if both parts of the foot are indicated by the letter F.
f) The abbreviation surf should be explained in the tables in Figure 9b for better legibility.
g) In figure 14c, please explain the abbreviation used for P. max.
Reviewer 4 Report
Comments and Suggestions for Authors
There has been many previous studies on foot biomechanics and insole design from 2004. This paper has to be very clear what new ideas it brings.
Previous model has considered soft tissue and bones, hence they have also considered foot dynamics. If this paper consider only soft tissue, this is a drawback - not an advantage, unless proved otherwise. Also the FE model is built from a no-load
some references include:
Cheung, J. T. M., Zhang, M., & An, K. N. (2004). Effects of plantar fascia stiffness on the biomechanical responses of the ankle–foot complex. Clinical Biomechanics, 19(8), 839-846.
Cheung, J. T. M., & Zhang, M. (2006, May). Finite element modeling of the human foot and footwear. In ABAQUS users’ conference (Vol. 20, No. 06, pp. 145-58).
Cheung, J. T. M., & Zhang, M. (2005). A 3-dimensional finite element model of the human foot and ankle for insole design. Archives of physical medicine and rehabilitation, 86(2), 353-358.
other comments
Section 2.2
"The methodology employed to reconstruct the foot biomodel has been recognized as 179 setting the guidelines in 3D biological tissue reconstruction [17,25]."
From the study it does not look like this methodology was employed. Example, acquisition of DICOM format. It is not clear which equipment was used and any data on the patient.
Discussion:
"The presented investigation stands out because it provides an innovative approach to analyzing the foot or any other biological tissue through the Finite Element Analysis since only soft tissues are considered"
The investigation does not stand out as many research has been done in this area for long time.
"It is in contrast to most biological numerical analyses that require the reconstruction of bone tissue to analyze soft tissue shock absorption behavior."
Need to have a strong data support to show that this method is better than previously developed method. Previous research has shown the need for bone and ligament structures.
Reviewer 5 Report
Comments and Suggestions for Authors
Nice designed Finite Element Analysis of foot pressure points through customization for orthotic devices.
I would recommended accepted after adding the limitation of this study.
Round 2
Reviewer 2 Report
Comments and Suggestions for Authors
I would like to thank the authors for their responses. Unfortunately, my concerns have not been addressed sufficiently for me to recommend the manuscript for publication; some even raised additional concerns (especially the responses to points 2 and 4). Considering that the only participant in the study had a notable "tendency to weight-bear a higher load to one side" due to their hip condition, it would be necessary to conduct additional testing on other (healthy) volunteers. Moreover, as the authors themselves acknowledge, "a second case study still needs to be developed since this methodology aims to replace experimental testing once the pressure prediction is in good agreement with medical methods." Should all the concerns raised be effectively addressed, the authors could consider resubmitting their manuscript for publication.
Reviewer 4 Report
Comments and Suggestions for Authors
The authors did not address my concerns sufficiently. hence I reject the paper in the current form.
Example:
1. "The equipment utilized was unnecessary to be described...".
As a reviewer, I feel it extremely important to describe all equipment, methods and procedures for scientific research. is not the normal research methods. So that the results can be properly replicated.
2. "In response to the first comments, as you pointed out, numerical analyses on the foot have been developed over the past years, establishing some of the methods applied for this manuscript. Based on other research manuscripts and the papers suggested in your part, the presented manuscript provides a different method to analyze pressure points in the plantar region."
If a new method is developed, then the new method has to be compared with other proposed methods that have been developed through the years. Previous methods have been validated and hence the new proposed method need to somehow prove that it is better. What is the scientific basis for providing the different method? The FE model is developed in low load condition, while the deformation is in loaded condition, there is bound to be more deformation at the loading areas. How does the new method plan to simulate dynamic situation, given that foot with bone structure have been able to simulate foot dynamic situation.